# ReTAG: Reasoning Aware Table to Analytic Text Generation

**Deepanway Ghosal**[1,2]   **Preksha Nema**[2]   **Aravindan Raghuveer**[2]

[1] DeCLaRe Lab, Singapore University of Technology and Design, Singapore
[2] Google Research, India
deepanway_ghosal@mymail.sutd.edu.sg, {preksh, araghuveer}@google.com

## Abstract

The task of table summarization involves generating text that both succinctly and accurately represents the table or a specific set of highlighted cells within a table. While significant progress has been made in table to text generation techniques, models still mostly generate descriptive summaries, which reiterates the information contained within the table in sentences. Through analysis of popular table to text benchmarks (ToTTo (Parikh et al., 2020) and InfoTabs (Gupta et al., 2020)) we observe that in order to generate the ideal summary, multiple types of reasoning is needed coupled with access to knowledge beyond the scope of the table. To address this gap, we propose **RETAG**, a table and reasoning aware model that uses vector-quantization to infuse different types of analytical reasoning into the output. **RETAG** achieves 2.2%, 2.9% improvement on the PARENT metric in the relevant slice of ToTTo and InfoTabs for the table to text generation task over state of the art baselines. Through human evaluation, we observe that output from RETAG is upto 12% more faithful and analytical compared to a strong table-aware model. To the best of our knowledge, RETAG is the first model that can controllably use multiple reasoning methods within a structure-aware sequence to sequence model to surpass state of the art performance in multiple table to text tasks. We extend (and open source 35.6K analytical, 55.9k descriptive instances) the ToTTo, InfoTabs datasets with the reasoning categories used in each reference sentences.

## 1 Introduction

In the task of Table to Text Generation (Yang et al., 2021), the output summaries usually are of two types: Descriptive and Analytical. A *descriptive* summary is formed with only information contained within the selected cells of the table and nothing else (Refer Table 1). On the other hand, an *analytical* summary is one that uses information

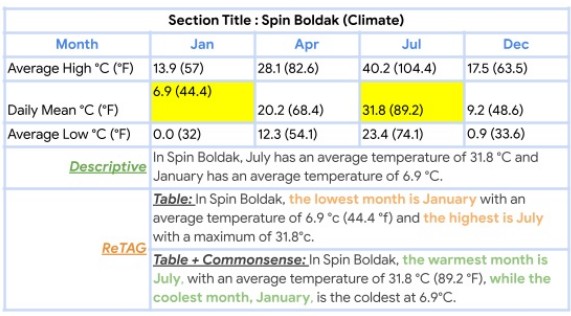

Figure 1: A descriptive summary simply states in words the numerical values highlighted in the cells. With RETAG we can produce different analytical summaries by controlling which reasoning categories are used: Tabular or (Tabular + Common Sense Reasoning)

beyond the selected cells and / or employs non-trivial reasoning. Being able to generate a high quality analytical summary is a critical property to improve the performance of Table to Text Models. We show later in our analysis that in ToTTo (Parikh et al., 2020) and Infotabs (Gupta et al., 2020), two widely used table to text benchmarks, roughly 27% and 44% of the reference summaries are analytical, respectively. In literature, there are five popular categories of reasoning used in analytical summarization: *1. Tabular Reasoning* where information in the table beyond the selected cells is used (Chen et al., 2020d; Saha et al., 2022; Zhao et al., 2022b; Chen et al., 2022), *2. Numerical Reasoning* that uses math operations on/across cells (Geva et al., 2020a; Dua et al., 2019; Amini et al., 2019), *3. Temporal Reasoning* that imparts knowledge about different units of time (Qin et al., 2021), *4. Commonsense Reasoning* to apply world knowledge (Talmor et al., 2019; Chen et al., 2020a) and *5. Entity Awareness* to distill knowledge about entities and their surface forms (Liu et al., 2018; Moiseev et al., 2022; Guu et al., 2020).

Most table to text models, use structure-aware architectures or pretraining strategies to internally equip the model with one or more types of analyt-

ical reasoning capabilities. We draw attention to two practical scenarios where reasoning categories should not be statically baked into a table to text model and more inference time control is needed.
**1. Dataset Diversity:** Tables like financial charts that consists of pre-dominantly numbers usually need numerical and temporal reasoning. On the other hand, tables like biographic information would need entity and common-sense knowledge more often. Since the same model would be used to summarize diverse tables, it becomes essential to be able to pick the appropriate reasoning categories based on the input table.

**2. Usage and Context Diversity:** Depending on the context of usage, a basketball match score-card table can be summarized in two distinct ways. For avid sports experts, temporal and tabular reasoning can be used to summarize interesting patterns in the table. On the other hand, for a newspaper article the focus would shift to entity knowledge and crisp tabular summary of match results.

Following from the examples above, we argue that explicit control is needed to dynamically pick the reasoning categories. during inference time. To the best of our knowledge, there is no prior literature on table to text summarization with explicit inference-time reasoning control.

Next, we study some key patterns in the human generated references in ToTTo (Parikh et al., 2020) and Infotabs (Gupta et al., 2020). The ToTTo and Infotabs validation set have 7700 and 196 tables with at most 3 reference summaries per table. We observe that close to 16.4% and 62.2% of the tables have summaries that use more than 2 reasoning categories simultaneously in ToTTo and Infotabs respectively. We term this as *Multi-Category Reasoning*. Such an example is shown in Figure 1. The actual table in the dataset lists the monthly temperatures recorded in a town *Spin Boldak* - we show only a sample here for illustration. We observe that to generate the ideal reference from the highlighted cells requires the model to posses two reasoning skills. First, *Tabular Summarization*: to compute the maximum and min values over the *daily mean* field. Second, *commonsense reasoning* to map the notion of maximum to warmest and that of minimum to coolest. Therefore, to attain human level accuracy on benchmarks like ToTTo and Infotabs a single generative model should have i) the knowledge of multiple reasoning skills ii) the ability to combine more than one skill to summarize

the highlighted cells. To the best of our knowledge, there is no prior generative table to text model that is capable of controlling and composing multiple reasoning categories.

In this paper, we make the following contributions to address the problems described above.

• We formulate a new summarization task called Reasoning Aware Table to Text motivated by failure situations that arise in real-world settings. To the best of our knowledge we are the first to propose this problem formulation. (Section 3)

• We present RETAG, a vector quantized approcah in encoder-decoder table to text models to generate rich analytical summaries with controllable multi-category reasoning. The use of RETAG improves the performance of state-of-the-art table to text models on the ToTTo and InfoTabs by 2.2%, 2.9%, respectively, in the PARENT metric. (Sections 4,5)

• We release an extended version of the ToTTo and the InfoTabs dataset that are annotated with the following popular reasoning categories: numerical, temporal, commonsense, table reasoning and entity knowledge as defined in (Gupta et al., 2020) (Section 6).

## 2 Related Work

Among recent advances Table To Text Models, two directions have led to significant improvement in analytical reasoning performance: *Table-Aware Architectures* and *Analytical Reasoning Pretraining*
**1. Table-Aware Architectures:** Novel model architectures (Liu et al., 2021; Gong et al., 2020; Xing and Wan, 2021; Li et al., 2021) have been proposed to efficiently encode tabular data and hence better use the implicit semantic structure of rows and columns. A large number of works have been introduced to incorporate table structure using pretraining strategies. (Herzig et al., 2020; Andrejczuk et al., 2022; Xing and Wan, 2021; Yin et al., 2020) introduce a pretraining strategy with specialized objectives for structured data: such as Masked Column Prediction(MCP), Adjacent Cell Prediction (ACP) and so on. Some of the above works, also use special row and column embedding to encode the table structure in the input. (Liu et al., 2021) learns table structure using the task of neural SQL execution over tables in their pretraining strategy. (Dong et al., 2022) presents an elaborate survey for table based pretraining strategies. It has been

shown that for analytical reasoning understanding table structure is a key ingredient. Therefore in our work we use TAPEX (Liu et al., 2021) as our backbone architecture.

**2. Analytical Reasoning Pretraining:** (Zhao et al., 2022b) introduces a new pretraining strategy with synthetic data augmentation using templates for very specific operations in numerical and temporal reasoning. (Zhu et al., 2021) also incorporates numerical and table operations in the model by appending symbolic reasoning operator selection on top of RoBerTa (Liu et al., 2019). (Zhao et al., 2022a; Chen et al., 2021) performs arithmetic reasoning through program generator module that converts the reasoning operation to an executable program to derive at the answer. Also, (Chen et al., 2022) disentangles computation with reasoning, by converting numerical reasoning aspect to an executable program. (Lei et al., 2022) models numerical question answering as expression tree generator task which helps in better alignment among question, table and paragraph.

To the best of our knowledge, none of existing techniques provide controllability with multi-category reasoning. This is the focus of our work. We discuss other prior literature on table to text datasets and controllability in Appendix G.

## 3 Problem Formulation

In this section, through a systematic human evaluation we first justify the choice of six reasoning categories (five analytical and the descriptive) that we introduce in Section 1. Next, we proceed to formally state the problem of Reasoning-Aware Table to Text Generation.

### 3.1 Analytical Reasoning Categories

InfoTabs (Gupta et al., 2020) was among the first to systematically present a comprehensive taxonomy for analytical reasoning with 14 categories. For the purposes of addressing data sparsity and non-overlap of reasoning categories we simplify the 14 categories proposed in (Gupta et al., 2020) into 6 categories as we explain below.

We sample roughly 20% of instances from the validation set of each dataset (1750 instances in ToTTo and 180 instances in InfoTabs respectively). The annotation below was done by a team of trained human experts and the details of the process is discussed in further detail in Section 6.

All instances were now annotated with the 14

| Reasoning Type | ToTTo | InfoTabs |
|---|---|---|
| Simple Lookup | 1543 | 74 |
| Numerical | 168 | 43 |
| Multirow | 118 | 14 |
| Temporal | 59 | 23 |
| Quantification | 12 | 36 |
| Entity Type | 36 | 9 |
| Commonsense | 72 | 31 |
| Lexical Reasoning | 11 | 28 |
| Named Entity | 3 | 2 |
| Coreference | 5 | 7 |
| Subjective | 5 | 4 |
| Syntactic Alterations | 5 | 20 |
| Ellipsis | 1 | 1 |
| Negation | 0 | 0 |

Table 1: Breakdown of the 14 categories on 1750 randomly sampled instances in ToTTo and 180 randomly sampled instances in InfoTabs.

categories as described in (Gupta et al., 2020). For (ToTTo, InfoTabs) the breakdown of the 14 categories are given in Table 1.

We observe that three categories (Ellipsis, Negation, Subjective) are very rare in both datasets. Due to the sparsity of training data in these categories, we do not consider them for the modeling task. The categories (Coreference, Lexical reasoning, Syntactic Alternations) are well covered by linguistic pre-training datasets and hence we do not explicitly try to model them either. We focus on the remaining 8 categories. Due to similarity of the categories and to coalesce training data, we combine (Numerical Reasoning, Quantification) into one category. Similarly we combine Entity Type and Named Entities into one category leaving us with a total of six categories.

Please note that the above methodology of constructing the six categories is an extension to (Gupta et al., 2020) and not a core contribution of our work. It is presented as a simple heuristic to form a viable set of non-overlapping, useful categories each with enough training data. The core contribution of our work is agnostic to the choice of actual categories themselves. Choosing a comprehensive and concise taxonomy for table to text reasoning is outside the scope of our work.

Informed by the above analysis, we annotated the entire dataset of InfoTabs with the six categories. The training dataset for ToTTo is large (120K samples), therefore we using a filtering heuristic (refer Appendix-A) to annotate a subset of the training set. The test set of ToTTo is not directly available as it is an online benchmark. Hence we annotated only the validation dataset for ToTTo.

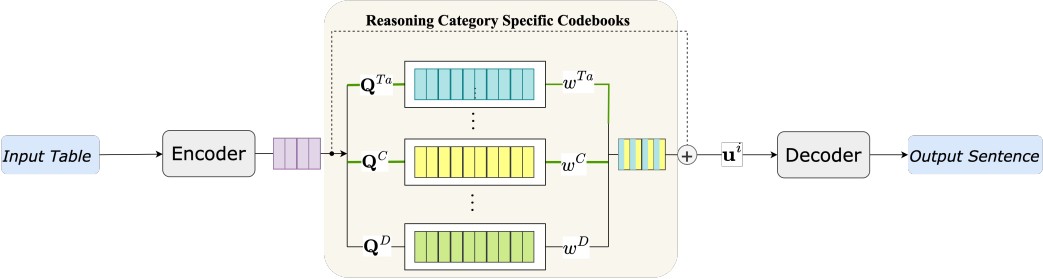

Figure 2: RETAG consists of three modules: Encoder, Vector Quantized Based Codebooks and Decoder. The output of the codebooks is element-wise added to the encoder output before being decoded.

Please note that no hyper-parameter tuning was done using the ToTTo validation set and was used for performance measurement alone.

### 3.2 Problem Statement

Let $\mathcal{R}$ be the set of six categories defined in Section 1: Descriptive, Tabular, Numerical, Temporal, Common-Sense and Entity reasoning. Given a table $\mathcal{T}$, set of cells $\{C_{ij}\}$ contained within $T$ and a reasoning category set $r \subseteq \mathcal{R}$, the task of reasoning aware table to text is to generate a summary that uses all the information contained in $\{C_{ij}\}$, applies analytical reasoning $r$ and then produces a truthful summary $\mathcal{S}$ confirming to $r$.

We use several automatic metrics to measure the relative quality of the generated sentences compared to the reference: BLEU (Papineni et al., 2002), ROUGE-L (Lin, 2004), semantic similarity with sentence embeddings (Reimers and Gurevych, 2019) using the MPNet model (Song et al., 2020); (ii) Between the tables and generated sentences: PARENT (Dhingra et al., 2019).

In addition we use three human eval metrics defined by the following questions:
**1.Reasoning:** Does the generated summary use all reasoning categories required in $r$?
**2. Faithful:** Is the generated summary hallucination free (*i.e.*, faithful to the table $\mathcal{T}$?)
**3. Coverage:** Are all cells in $\{C_{ij}\}$ used in the generated summary?

## 4 Proposed Model

We propose our model RETAG for controllable multi-skill reasoning table to text generation. We embed the two key aspects required to generate analytical sentences: control on reasoning categories, and being reasoning aware. First, to better model each reasoning category with control, we use a vector quantization (VQ) strategy in RETAG (§4.2 and §4.4). Second, to precisely model each reasoning

category, we use a pretraining strategy to better learn the reasoning categories from structured tables and free-form text data (§4.3).

For any table-to-text model, a basic property required is the efficient understanding of table structure. To infuse this aspect, we use the pretrained TAPEX model (Liu et al., 2021) as the encoder-decoder backbone in our framework. TAPEX is a BART (Lewis et al., 2020) model fine-tuned on the task of neural SQL execution over tables. Authors show that this is an effective pretraining strategy for various table related downstream tasks.

It is important to note that our model architecture contributions are not limited to TAPEX, and thus we integrate the same modules in T5 (Raffel et al., 2020a) based models and benchmark their performance later in the paper.

### 4.1 Preliminaries

We denote the encoder and decoder as $\mathbf{E}$ and $\mathbf{D}$. We use the question $q$ to pass the reasoning tags in the input: *Generate a sentence with **TAG reasoning** based on the following table?* for analytical generation; *Generate a **descriptive** sentence based on the following table?* for the descriptive generation task. We concatenate $q$ with the linearized table $t$ to form the input $\mathbf{x}$. The encoded vector is $\mathbf{E}(\mathbf{x}) \in \mathcal{R}^{N \times H}$, where $N$ is the number of tokens in $\mathbf{x}$, and $H$ is the latent dimension. For our base model, the decoder $\mathbf{D}$ generates the corresponding output sentence $\mathbf{y} = \mathbf{D}(\mathbf{E}(\mathbf{x}))$.

### 4.2 Vector Quantized Reasoning

Our primary objective is to incorporate reasoning level control in our model RETAG. However, one of the main challenges of generating analytical sentences is to learn category specific aspects, which can be used to perform interaction between these categories for complex reasoning. To achieve this, we sandwich a vector quantization module between

the encoder and the decoder. Each reasoning category has its own *codebook* on which the vector quantization operation is performed.

We use the encoded representation $\mathbf{E}(\mathbf{x})$ to intervene reasoning specific knowledge from codebooks to create a new reasoning aware representation which is then passed to the decoder $\mathbf{D}$.

A codebook is denoted as $\mathbf{c} \in \mathcal{R}^{K \times H}$, which is a latent embedding space of size $K$, with $H$ being the dimension of each of the codes $c_i$ in $\mathbf{c}$ (Van Den Oord et al., 2017). We have a codebook $\mathbf{c}^r$ for each of the five reasoning categories $r$. For completeness, we also have a separate sixth codebook for the descriptive category. We use the codebooks to extract reasoning based representation for the encoded $\mathbf{E}(x)$ using quantization process $\mathbf{Q}$. It maps each token vector in $\mathbf{E}(\mathbf{x})$ to the nearest code $c_k^r$ for the given category codebook $\mathbf{c}^r$:

$$\mathbf{Q}^r(\mathbf{E}(x)_n) = c_k^r, \ \forall \ n = 1, \dots, N$$
$$\text{where, } k = \operatorname{argmin}_j ||\mathbf{E}(x)_n - c_j^r||_2 \tag{1}$$

Now, to model multi-category reasoning for generating analytical sentences, we propose the following weighted summation technique:

$$\mathbf{Q}^r(x) = \mathbf{1}_R \sum_{i=1}^R w^i . \mathbf{Q}^i(x) \tag{2}$$

Here, $w^i$ represents scalar weights for each reasoning category, predicted from the last layer of the encoder $\mathbf{E}$, through an additional head layer. The binary labels $\mathbf{1}_R$ simplify the equation so that the codebooks used are restricted only to the specified reasoning labels. Furthermore, we add a residual connection between the reasoning based representation and the original encoder representation. We then pass the resultant vector to the decoder $\mathbf{D}$ to generate the analytical sentence $\mathbf{y}^a$.

$$\mathbf{u}^a = \mathbf{Q}^r(\mathbf{x}) + \mathbf{E}(\mathbf{x})$$
$$\mathbf{y}^a = \mathbf{D}(\mathbf{u}^a) \tag{3}$$

We also generate the descriptive sentence $\mathbf{y}^d$ in a similar manner using a residual connection between the encoded vector and the quantized representation from the descriptive codebook. For ease of understanding, we refer to $\mathbf{Q}^r(\mathbf{x})$ as $\mathbf{Q}(\mathbf{x})$.

## 4.3 Reasoning-Aware Pretraining

In order to generate analytical sentences with our proposed architecture, it is crucial that the codebooks are rich in representing each of the reasoning categories efficiently. The five reasoning categories we use extends beyond performing inferences on specific tables. Therefore, we explore pretraining strategies with various free-form and structured-data based datasets having the specific reasoning components. We collect the following datasets: (i) numerical and textual data (ND, TD) from Geva et al. (2020b), DROP (Dua et al., 2019), MathQA (Amini et al., 2019) for numerical reasoning, (ii) LogicNLG (Chen et al., 2020b), Logic2Text (Chen et al., 2020e) for numerical reasoning and table knowledge, (iii) WIKIBIO (Liu et al., 2018) for table reasoning and entity knowledge, (iv) CommonsenseQA (Talmor et al., 2019), PIQA (Bisk et al., 2020), Social-IQA (Sap et al., 2019) for commonsense reasoning, and (v) MC-TACO (Zhou et al., 2019) for temporal reasoning.

We have a total of 276k instances from the above datasets spanning over the five reasoning categories. We formulate a seq-to-seq text generation task and pretrain our model (encoder, decoder, codebook) on the reasoning-aware dataset. We detail the model training strategy in Section 4.4.

## 4.4 Reasoning Control based Fine-tuning

To further improve upon reasoning based representations, we add a classifier network $\mathbf{M}$ on top of the residual features $\mathbf{u}^a$ and $\mathbf{u}^d$, which classifies it into *analytical* and *descriptive* classes. This classification constraint helps the model to broadly learn the difference between analytical and descriptive sentence. We term this strategy as CI (Classification of Intermediate activations). We show later that this classification strategy helps in improved generation for both descriptive and analytical sentences. The overall loss function for a batch consisting of both analytical and descriptive references, is as follows:

$$L = -[(\hat{\mathbf{y}}^d * \log \mathbf{y}^d + \hat{\mathbf{y}}^a * \log \mathbf{y}^a)]_{(1)}$$
$$- [\log(\mathbf{M}(\mathbf{u}^d)^-) + \log(\mathbf{M}(\mathbf{u}^a)^+)]_{(2)}$$
$$+ [||sg[\mathbf{E}(\mathbf{x})] - \bar{\mathbf{Q}}(\mathbf{x})||_2^2]_{(3)} + [\beta||\mathbf{E}(\mathbf{x}) - sg[\bar{\mathbf{Q}}(\mathbf{x})]||_2^2]_{(4)}$$

The loss function consists of four components: **Generative Loss**: The first term is the cross-entropy loss over the vocabulary for generating the gold descriptive and analytical sentences $\hat{\mathbf{y}}^d$ and $\hat{\mathbf{y}}^i$. **Classifier Loss**: The second term is the cross-entropy loss from the classification constraint. We denote $^-$ and $^+$ as the descriptive and analytical class. **Codebook Loss**: stop-gradient ($sg$) in the third term is required for training the codebooks as the quantization process with *argmin* operation in Equation (1) is non-differentiable. Van

| Dataset | Model | Strategy | Overall Performance | | | | | Analytical Performance | | | | | Descriptive Performance | | | | |
|---|---|---|---|---|---|---|---|---|---|---|---|---|---|---|---|---|---|
| | | | B1 | B4 | R-L | Sim | PAR | B1 | B4 | ROUGE | Sim | PAR | B1 | B4 | ROUGE | Sim | PAR |
| ToTTo | T5 | No Tags | 68.91 | 22.78 | 59.24 | 86.51 | 63.12 | 65.60 | 19.45 | 53.62 | 84.12 | 60.11 | 70.18 | 24.12 | 61.24 | 87.66 | 66.15 |
| | FLAN T5 | | 69.83 | 22.67 | 59.69 | 87.17 | 63.31 | 65.47 | 19.77 | 54.36 | 84.51 | 60.05 | 71.09 | 24.94 | **61.61** | 88.13 | 66.27 |
| | TAPEX | | 69.92 | 22.57 | 59.18 | 87.13 | 63.51 | 65.71 | 18.60 | 53.33 | 84.29 | 60.34 | 71.20 | 24.26 | 61.28 | 88.16 | 66.20 |
| | T5 | Tags | 69.18 | 22.77 | 59.21 | 87.11 | 64.12 | 66.05 | 19.77 | 54.00 | 84.66 | 61.46 | 70.51 | 24.08 | 61.08 | 87.99 | 65.99 |
| | FLAN T5 | | 70.37 | **23.62** | 59.70 | 87.11 | 64.41 | 66.88 | **20.33** | **54.62** | **84.69** | 61.25 | 71.84 | **25.05** | 61.52 | 87.99 | 66.55 |
| | TAPEX | | 70.79 | 22.63 | 59.07 | 86.93 | 64.91 | 66.82 | 18.25 | 52.91 | 83.89 | 61.76 | 72.43 | 24.49 | 61.29 | 88.03 | 66.89 |
| | T5 | ReTAG | 70.05 | 22.34 | 59.01 | **87.27** | 64.65 | 66.43 | 18.57 | 53.15 | 84.19 | 61.86 | 71.54 | 23.94 | 61.12 | **88.31** | 66.75 |
| | FLAN T5 | | 70.46 | 22.56 | 59.10 | 86.99 | 64.85 | 66.91 | 18.86 | 53.06 | 84.00 | 62.01 | 71.90 | 24.12 | 61.27 | 88.07 | 67.12 |
| | TAPEX | | **71.24** | 23.03 | **60.20** | 86.81 | **65.22** | **67.90** | 19.04 | 53.08 | 83.84 | **62.57** | **72.59** | 24.69 | 61.14 | 87.88 | **67.32** |
| InfoTabs | T5 | No Tags | 40.64 | 17.22 | 37.26 | 65.42 | 23.26 | 36.44 | 4.60 | 32.63 | 65.22 | 13.04 | 42.54 | **22.17** | 40.97 | **65.58** | 33.30 |
| | FLAN T5 | | 39.43 | 17.00 | 36.32 | 65.09 | 24.69 | 35.61 | 4.01 | 31.17 | 65.13 | 11.72 | 41.02 | 21.62 | 40.47 | 65.06 | 32.12 |
| | TAPEX | | 44.04 | 16.82 | 36.11 | 63.11 | 25.01 | 42.01 | 5.45 | 32.79 | 62.69 | 14.86 | 44.99 | 21.24 | 38.77 | 63.46 | 32.20 |
| | T5 | Tags | 41.54 | 17.34 | **38.90** | 66.55 | 25.75 | 43.41 | **7.85** | **38.12** | **69.06** | 17.28 | 40.77 | 20.61 | 39.52 | 64.52 | 32.56 |
| | FLAN T5 | | 39.82 | 17.02 | 38.54 | **66.74** | 26.77 | 38.80 | 6.47 | 37.05 | 68.40 | 16.90 | 40.25 | 20.98 | 39.74 | 65.40 | 34.71 |
| | TAPEX | | 44.60 | 17.28 | 36.96 | 64.13 | 25.52 | 41.67 | 6.16 | 33.97 | 64.36 | 16.92 | 46.07 | 22.10 | 39.37 | 63.94 | 32.31 |
| | T5 | ReTAG | 42.60 | 17.28 | 36.96 | 64.13 | 26.58 | 39.67 | 6.16 | 33.97 | 64.36 | 16.40 | 44.07 | 22.10 | 39.37 | 63.94 | 34.78 |
| | FLAN T5 | | 39.93 | 11.96 | 32.61 | 62.03 | 25.63 | 30.64 | 3.12 | 27.76 | 60.55 | 15.48 | 46.55 | 18.02 | 36.52 | 63.22 | 33.96 |
| | TAPEX | | **45.95** | **17.71** | 37.32 | 64.91 | **26.97** | **45.01** | 6.80 | 34.46 | 65.52 | **17.75** | 46.38 | 21.95 | 39.62 | 64.42 | **35.05** |

Table 2: Automatic evaluation results for analytical and descriptive sentence generation with in the Infotabs and ToTTo datasets. Sim denotes the cosine similarity from a sentence embedding model. B1, B4, R-L, PAR denotes BLEU1, BLEU4, ROUGE-L, PARENT scores respectively. Reasonig awareness (using TAGS or RETAG ) consistently improves the performance of all the models.

Den Oord et al. (2017) used a straight-through estimator to approximate the gradient by copying the gradient from the decoder input to the encoder output. We use the common term $\bar{\mathbf{Q}}(\mathbf{x})$ to represent the quantized vector for both analytical, descriptive instances. **Commitment Loss**: The fourth term scaled by hyperparameter $\beta$ helps the encoder output to commit to the closest codebook vector. For codebook pretraining in Section 4.3, we use the loss terms (3), (4) in the first stage, and then (1), (3), (4) in the second stage.

## 5 Experiments

We evaluate the performance of RETAG as follows: (i) Performance comparison against strong baselines (ii) Ablation study of design choices used in RETAG (iii) Effectiveness of Vector Quantization (iv) Reasoning category wise quantitative analysis and (v) Human evaluations for faithfulness and reasoning control. Since our primary contribution is on multi-category reasoning, we benchmark RETAG 's performance for ToTTo valid and InfoTabs test datasets which have heterogeneous reasoning categories. Hence, we do not evaluate against datasets that are specific to one type of reasoning, such as LogicNLG, Numerical-NLG, etc.

We use the following notations for our experiments: given a question $q$ with a linearized table $t$, we concatenate them to form the input $\mathbf{x}$ (as mentioned in Section 4.1). The table may contain some highlighted cells, which can be enclosed within special indicators such as *<hl>* or can be addition-

ally mentioned at the beginning or end of the table string. We assume that the highlighted cells would be indicated in either of these ways within the linearized table $t$. We use the strategy devised in Liu et al. (2021) for linearizing the table. Additionally we are also given the reasoning categories $r$ and the corresponding output sentence $\mathbf{y}$.

### 5.1 Main Results

We use the following models in our experiments: T5-Large (768M parameters) (Raffel et al., 2020b), FLAN T5-Large (768M parameters) (Chung et al., 2022) and TAPEX (406M parameters) (Liu et al., 2021). We use the models in three different ways:

1. We use only $\mathbf{x}$ to directly generate $\mathbf{y}$. No information about $r$ is consumed by the model. This is the usual seq-to-seq baseline. We denote this strategy as No Tags in Table 2.

2. We use information about $r$ as part of the question $q$. Then we train the models to generate $\mathbf{y}$ from $\mathbf{x}$. The category information is thus used as part of the input string $x$. We deonte this as Tags in Table 2.

3. We use information about $r$ with the codebook selection strategy. This is our proposed RETAG method with pretraining as mentioned earlier in Section 4 and Section 4.3.

**ToTTo**: We observe that the models with the TAGS or RETAG approach generally outperform the models with No Tags. We observe this result

| #CBs | CI Loss | ToTTo | | | InfoTabs | | |
|---|---|---|---|---|---|---|---|
| | | B1 | B4 | R-L | B1 | B4 | R-L |
| 2 | No | 68.17 | 20.89 | 56.78 | 43.87 | 16.09 | 35.64 |
| 2 | Yes | 68.75 | 21.54 | 57.30 | 44.25 | 16.52 | 35.71 |
| 6 | No | 70.54 | 22.71 | 58.49 | 45.38 | 17.32 | 37.26 |
| 6 | Yes | **71.24** | **23.03** | **59.00** | **45.95** | **17.71** | **37.32** |

Table 3: Results on the overall ToTTo validation and In-foTabs test set with different number of codebooks (CBs) and the classifier loss for the TAPEX RETAG model.

across most of the evaluation metrics. In particular the TAPEX RETAG model achieves around 1% improvement in BLEU-1, ROUGE-L and around 2% improvement over the NO TAGS models for the overall performance in the validation set. The performance improvement in the analytical set is slightly more prominent compared to the descriptive set in the BLEU-1 and PARENT metrics. We postulate that the tag-based distinction between analytical and descriptive control improves the performance for descriptive sentences, as the model gets a clear signal of when to describe the content versus when to reason.

We further study the importance of augmenting the input table with reasoning categories for fine-tuning the models in the Tags group of results. We observe that it leads to increment in performance across the BLEU-1 and PARENT metrics for the overall set. In the analytical set, the improvement in performance is around 1% across for BLEU-1 and PARENT for all the models.

**InfoTabs** We achieve considerable improvement with the TAGS and RETAG approach for over-all performance and analytical set performance in Infotabs. TAPEX model has superior performance over the T5 family model in NO TAGS as the TAPEX is trained on table corpora on table understanding tasks. However, the performance of the comparatively poorer T5 and FLAN-T5 model is significantly improved with the use of categorical information. It re-iterates the importance of adding reasoning based control in various models. Our proposed TAPEX RETAG model still outperforms the TAPEX TAG model by more than 1% for BLEU-1 and PARENT for overall set, and around 3% and 1% for BLEU and PARENT for the analytical set.

## 5.2 Ablation Studies

RETAG consists of three main components: the codebooks, the classification objective to differentiate analytical and descriptive and the pretraining technique of the codebooks. In this section, we study the effect of these three components on the

| Pretraining | ToTTo | | | InfoTabs | | |
|---|---|---|---|---|---|---|
| | B1 | B4 | R-L | B1 | B4 | R-L |
| No Pretraining | 69.86 | 21.66 | 58.25 | 44.49 | 16.75 | 36.13 |
| With Pretraining | **71.24** | **23.03** | **59.00** | **45.95** | **17.71** | **37.32** |

Table 4: Results on the overall ToTTo validation and In-foTabs test set with different pretraining strategies for the TAPEX RETAG model.

performance.

**1. Number of Codebooks** In Table 3, we present the performance of RETAG for two and six code-book setup with category specific quantization. The six codebook setup is the model we originally introduced in Section 4. We now combine the five reasoning categories into one analytical category and keep the descriptive category as the other category. This results in the two codebook setup, which we then benchmark with the TAPEX RETAG model.

We observe that six codebook setup consistently outperforms two codebook setup across the various metrics for both ToTTo and InfoTabs datasets (first and third rows in Table 3). We conclude that category specific codebooks provide more flexibility and capacity in the model with better control over the generations.

**2. Intermediate Activation Classification** In Table 3 we also study the effect of classifying the residual features $\mathbf{u}^a$ and $\mathbf{u}^d$ into analytical and descriptive classes with the CI classification loss (Section 4.4). We observe that the RETAG performance improves consistently with CI constraint for both the two and six codebook setups on both the ToTTo and InfoTabs dataset.

**3. Codebook Pretraining Strategies** In Table 4, we study the efficacy of the codebook pretraining stratey. All the RETAG models here have six codebooks and the CI constraint enabled. As the name suggests, in the no pretraining strategy, we do not perform any pretraining on the model and directly fine-tune it for table-to-text task. In the with pretraining strategy, we start with random initialization of the codebooks and pretrain the codebooks along with the encoder and decoder of the models on the ensemble of reasoning dataset mentioned earlier in Section 4.3. We found that the the pretraining strategy is considerably more effective for the final tasks in ToTTo and InfoTabs across all the evaluation metrics, as reported in Table 4.

## 5.3 Codebook Analysis

In this section, we analyze the effectiveness of the codebooks for analytical generations as follows.

| Dataset | Model | Strategy | Numerical | | Commonsense | | Temporal | | Table | | Entity | | 2 Category | | 3 Category | |
|---|---|---|---|---|---|---|---|---|---|---|---|---|---|---|---|---|
| | | | B1 | R-L | B1 | R-L | B1 | R-L | B1 | R-L | B1 | R-L | B1 | R-L | B1 | R-L |
| ToTTo | TAPEX | No Tags | 64.73 | 49.10 | 67.81 | 54.21 | 69.63 | 58.29 | 65.67 | 50.86 | 64.97 | 50.73 | 65.83 | 51.17 | 65.31 | 47.12 |
| | | Tags | 65.12 | 49.75 | 68.25 | **55.05** | 69.78 | **58.98** | 65.94 | **51.41** | 65.05 | 50.70 | 66.28 | **51.70** | 65.08 | 47.72 |
| | | ReTag | **66.28** | **49.96** | **68.47** | 54.75 | **70.77** | **58.98** | **67.20** | 51.31 | **67.10** | 50.95 | **67.57** | 51.36 | **65.47** | 48.03 |
| InfoTabs | TAPEX | No Tags | 40.47 | 29.99 | 43.71 | 34.64 | 41.63 | 31.38 | 40.63 | 30.63 | 41.80 | 32.64 | 39.22 | 28.94 | 40.72 | 30.37 |
| | | Tags | 42.35 | **30.93** | 45.80 | 36.01 | 43.11 | 30.73 | 44.95 | 33.27 | 43.52 | 34.25 | 42.24 | 31.38 | **45.76** | **33.52** |
| | | ReTag | **42.90** | 30.78 | **46.88** | **36.45** | **44.47** | **31.43** | **45.10** | **33.36** | **44.63** | **35.26** | **43.39** | **32.39** | 45.13 | 33.18 |

Table 5: Category-wise automatic evaluation results. We observe improvement in performance for each category and combination of categories after the introduction of category specific information with the TAGS and RETAG approach.

| Label Type | ToTTo | | | | | | Infotabs | | | | | |
|---|---|---|---|---|---|---|---|---|---|---|---|---|
| | 1-Category | | 2-Category | | 3-Category | | 1-Category | | 2-Category | | 3-Category | |
| | B1 | R-L | B1 | R-L | B1 | R-L | B1 | R-L | B1 | R-L | B1 | R-L |
| Random | 65.16 | 50.93 | 64.28 | 50.05 | 32.23 | 46.37 | 40.29 | 33.61 | 39.51 | 29.01 | 39.38 | 30.57 |
| Gold | **68.26** | **54.37** | **67.57** | **51.36** | **65.47** | **48.03** | **45.51** | **35.76** | **43.39** | **32.39** | **45.13** | **33.18** |

Table 6: Effectiveness of the codebooks for generating analytical sentences with the TAPEX RETAG model. Choosing random analytical labels during inference leads to drop in performance, showing that the codebooks learn meaningful representations of the reasoning categories.

1. **Category-Wise Performance** We evaluate results across reasoning categories in Table 5. We pick out instances having just a single category annotated and report the results for them in the L.H.S of Table 5. It helps us in analyzing the effect on each category without the involvement of others. We note that the six codebooks in TAPEX RETAG helps improve performance across all the five reasoning categories in comparison to the baseline model. The TAPEX model with TAGS also helps in improving the performance over the TAPEX model without any tags.

2. **Multi-Category Reasoning**: We also study TAPEX RETAG for complex analytical sentences that involves two or more reasoning categories. We report average results for instances having two or three categories in the R.H.S of Table 5. TAPEX RETAG consistently outperforms the baseline by 2% in BLEU1 score for ToTTo and around 4% in BLEU1 and 3% in ROUGE-L for the InfoTabs dataset.

3. **Random Reasoning Labels** In Table 6, we study the performance when random reasoning categories are sent to the model during inference. We observe that, the evaluation scores drop by significant margins across sentences with single and multiple categories across both the datasets. This shows, that the codebook encodes meaningful and distinct representations for each reasoning category. We show in Appendix F that RETAG is also able to beat SOTA baselines for a Table QA task on a dataset called Turning Tables (Yoran et al., 2022) where the reasoning categories are provided.

We conclude that RETAG models capture reasoning-specific information in each codebook through pretraining, which it uses effectively for both single and multi-category analytical sentence generation.

## 5.4 Human Evaluation

We sample 500 instances from the ToTTo validation set and generate their corresponding analytical sentences from four different models specified in Table 7. Given an instance of table and the reasoning categories, we ask the annotators to evaluate the the sentence on three questions as described in Section 3.2: Reasoning, Faithful and Coverage.

We ask the annotators to provide a label between – *yes* (score 1), *partially* (score 0.5), or *no* (score 0). We compare TAPEX RETAG against the TAPEX model without tags for human evaluation to quantify faithfulness and reasoning control. We collect 3 ratings for each sample and compile the majority label results in percentage scale in Table 7. We observe that with explicit reasoning control, TAPEX RETAG generates 13% more faithful and 12% better coverage on the reasoning categories on analytical sentences as compared to TAPEX No Tags. We found very good inter rater agreement for the human evaluation task. The Fleiss' kappa score between the three annotators for human evaluation were as follows: 0.5011 on the reasoning categories, 0.5583 on the faithfulness, and 0.6722 on the coverage of highlighted cells. Some examples of RETAG output can be found in Appendix E.

| Model | Human Evaluation | | |
|---|---|---|---|
| | Reasoning | Faithful | Coverage |
| TAPEX NO TAGS | 51.0 | 76.0 | 69.4 |
| TAPEX RETAG | **63.2** | 89.3 | 77.6 |
| Two Codebooks | 53.9 | 90.6 | 78.9 |
| No Pre-training | 61.1 | **91.3** | **79.9** |

Table 7: Human evaluation results for analytical sentence generation in ToTTo. Scores are shown in % scale.

## 6 Reasoning Category Tagged Dataset Release

As explained in Section 3.1, we will release 31.4K analytical instances and 50.6K for the ToTTo train and validation set. We will also release 4.2K analytical instances and 5.3K descriptive sentences over the entire InfoTabs dataset. This section explains the human labeling methodology and the corresponding performance metrics.

We prepared detailed annotation instructions, qualifying questions and trained a pool of 14 crowdsource annotators. The annotators are based in India and are fluent in English. The annotators were paid at rates above required by local wage laws.

We instructed the annotators to choose one or more of the five reasoning categories for analytical sentences. We instructed them to keep the five reasoning categories and the *Descriptive* category exclusive i.e. a sentence is descriptive only when it does not use any of the other five reasoning categories. Three annotators labeled every instance and we keep only those label voted by atleast two raters. The annotators reached a high consensus agreement on the task. 86.81% of ratings had all three raters agree on the binary class for categorizing between descriptive and analytical. 75.12% all three raters agreed on the exact same set of categories for choosing the analytical categories.

## 7 Conclusion

In this paper, we presented the case for Reasoning-aware table-to-text models. We introduced RETAG, a vector quantized approach for encoder-decoder table-to-text models with explicit control across reasoning categories. RETAG beats SOTA models for ToTTo and InfoTabs datasets for analytical and descriptive sentence generation. We will also release close to 35.6K instances of reasoning category tagged analytical abd 55.9k instances of descriptive table to text data.

## 8 Limitations

Some of the limitations of our work are as follows. **First**, the dataset curation and performance evaluation was restricted to datasets in the English language, and does not extend to non-dominant languages. **Second**, several advanced methods have been introduced for numerical reasoning. Our current strategy to incorporate reasoning is data-centric. However, we would like to emphasize that the explicit reasoning control is complementary to the existing methods and in future works, advance methods to infuse reasoning can be used alongside our method. **Third**, to gain explicit reasoning control for newer domain/reasoning category, involves few examples to be annotated to bootstrap the model using our method. **Fourth**, although RETAG is designed for multiple skill reasoning, in future work we will also benchmark RETAG against reasoning specific datasets such as Logic-NLG.

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

## A  Dataset Details and Filtering Strategy

**ToTTo** (Parikh et al., 2020) is an open domain table-to-text dataset where the task is as follows: given a table with a set of highlighted cells, generate a sentence constrained upon the highlighted cells. The tables are collected from Wikipedia and metadata information (table title, section title containing the table) are available. We observe that many reference sentences in the dataset are *analytical*, as they are based on different forms of reasoning, as opposed to simply stating the highlighted cells verbatim. We compute a fuzzy string match between the annotated reference and the highlighted cells + metadata of the table. We identify ~63k instances (out of total 120k in ToTTo training set) using various constraints: (i) reference contain non-stop words not present in the table; (ii) fuzzy match $< 80\%$, (iii) fuzzy match $< 85\%$ and presence of superlatives, comparatives, or numerics (largest, faster, third, etc.). We annotate a small pilot set and surmised that $\sim 40\%$ of the 63k instances are analytical and these 63k instances would contain most of the analytical instances in ToTTo training set. In addition we also use the full validation set for annotation, as mentioned previously in Seciton 6.

**InfoTabs** (Gupta et al., 2020) is a dataset for table natural language inference (NLI), where the tables (collected from Wikipedia info-boxes) are considered as semi-structured premise and human written sentences are provided as hypothesis. The task is predict whether the sentence is an entailment, contradiction, or neutral w.r.t the table. We compute the fuzzy match between the linearized table and the entailment instances and identify: (i) 4.9k instances with fuzzy match $< 75\%$ as potentially analytical, and (ii) 1.7k instances with fuzzy match $> 80\%$ as potentially non-analytical. To keep the the two categories somewhat balanced, we additionally generate 2.9k potentially non-analytical instances using a keyword to sentence seq2seq model trained on the WikiTableText dataset (Bao et al., 2018). In total, we have 9.5k instances to annotate.

## B Reasoning Categories

We show some examples of table, sentence pairs and the corresponding reasoning categories Figure 3 - 7. Some of these examples show how multiple reasoning categories could be combined to generate an analytical sentence. We also show an example of descriptive sentence in Figure 8.

**Eşref Apak**

**Section Title**: Achievements
**Table Section Text**: *None*

| Year | Competition | Venue | Position | Notes |
|------|-------------|-------|----------|-------|
| **Representing Turkey** | | | | |
| 2000 | World Junior Championships | Santiago, Chile | 1st | 69.97 m NJR |
| 2001 | Mediterranean Games | Tunis, Tunisia | 6th | 71.06 m |
| 2003 | European U23 Championships | Bydgoszcz, Poland | 2nd | 76.52 m |
| 2004 | Olympic Games | Athens, Greece | 3rd | 79.51 m |
| | Universiade | Izmir, Turkey | 2nd | 76.18 m |
| 2005 | Mediterranean Games | Almería, Spain | 1st | 77.88 m |
| | World Championships | Helsinki, Finland | 17th (q) | 73.04 m |
| 2006 | European Championships | Gothenburg, Sweden | 19th (q) | 70.17 m |
| 2007 | World Championships | Osaka, Japan | 11th | 76.59 m |
| 2008 | Olympic Games | Beijing, China | 16th (q) | 74.45 m |
| 2009 | World Championships | Berlin, Germany | 27th (q) | 70.70 m |
| 2012 | Olympic Games | London, United Kingdom | 17th (q) | 73.47 m |
| 2015 | World Championships | Beijing, China | 17th (q) | 73.01 m |
| 2016 | European Championships | Amsterdam, Netherlands | – | NM |
| | Olympic Games | Rio de Janeiro, Brazil | 24th (q) | 70.08 m |
| | Islamic Solidarity Games | Baku, Azerbaijan | 1st | 74.32 m |
| 2017 | European Team Championships | Lille, France | 2nd | 71.53 m |
| | World Championships | London, United Kingdom | 16th (q) | 73.55 m |
| 2018 | European Championships | Berlin, Germany | 17th (q) | 72.70 m |

**Sentence**

Apak participated in the 2005 Mediterranean Games in Almería and won the gold medal with his throw of 77.88 metres.

Figure 3: An example of commonsense reasoning in the above table.

**Swaziland at the 1992 Winter Olympics**

**Section Title**: Alpine skiing
**Table Section Text**: *None*

| Athlete | Event | Race 1 Time | Race 2 Time | Total Time | Rank |
|---------|-------|-------------|-------------|------------|------|
| | Men's Super-G | | | 1:29.39 | 79 |
| Keith Fraser | Men's giant slalom | 1:21.93 | 1:19.83 | 2:41.76 | 63 |
| | Men's slalom | DNF | - | DNF | - |

**Sentence**

Keith Fraser finished the second run in a faster time of 1 minute and 19.83 seconds, which led him to the 63rd position.

Figure 4: An example of numerical reasoning in the above table.

## C Experimental Setup and Computational Resources

We use beam search to generate outputs from the our generative models. We used a beam length of 10 is used. All models were trained with the AdamW optimizer with a learning rates of 1e-6,

**List of VFL/AFL records**

**Section Title**: Most career games
**Table Section Text**: *None*

| Rank | Games | Player | Club | Career span |
|------|-------|--------|------|-------------|
| 1 | 432 | Brent Harvey | North Melbourne | 1996-2016 |
| 2 | 426 | Michael Tuck | Hawthorn | 1972-1991 |
| 3 | 403 | Kevin Bartlett | Richmond | 1965-1983 |
| 4 | 400 | Dustin Fletcher | Essendon | 1993-2015 |
| 5 | 383 | Robert Harvey | St Kilda | 1988-2008 |

**Sentence**

Michael Tuck held the record as the VFL/AFL games record holder with 426 games until 2016 when the record was broken by North Melbourne's Brent Harvey.

Figure 5: An example of temporal reasoning in the above table.

**Busch Stadium**

**Section Title**: Professional soccer
**Table Section Text**: *None*

| Date | Winning Team | Result | Losing Team | Tournament | Spectators |
|------|--------------|--------|-------------|------------|------------|
| May 23, 2013 | England Manchester City | 4-3 | England Chelsea | Club Friendly | 48,263 |
| November 18, 2013 | Argentina | 2-0 | Bosnia and Herzegovina | International Friendly | 30,397 |
| April 4, 2015 | United States women | 4-0 | New Zealand women | Women's International Friendly | 35,817 |
| November 13, 2015 | United States | 6-1 | Saint Vincent and the Grenadines | 2018 FIFA World Cup qualification | 43,433 |
| August 1, 2016 | Italy Roma | 2-1 | England Liverpool | Club Friendly | 29,000 |

**Sentence**

The highest attendance for a sports event in Busch Stadium was on May 23, 2013, when 48,263 people watched Chelsea and Manchester City play a friendly match.

Figure 6: An example of numerical and table reasoning in the above table.

**Francesco Bagnaia**

**Section Title**: By season
**Table Section Text**: *None*

| Season | Class | Motorcycle | Team | Number | Race | Win | Podium | Pole | FLap | Pts | Plcd |
|--------|-------|------------|------|--------|------|-----|--------|------|------|-----|------|
| 2013 | Moto3 | FTR Honda | San Carlo Team Italia | 4 | 17 | 0 | 0 | 0 | 0 | 0 | NC |
| 2014 | Moto3 | KTM | SKY Racing Team VR46 | 21 | 16 | 0 | 0 | 0 | 1 | 50 | 16th |
| 2015 | Moto3 | Mahindra | MAPFRE Team MAHINDRA Moto3 | 21 | 18 | 0 | 1 | 0 | 1 | 76 | 14th |
| 2016 | Moto3 | Mahindra | Pull & Bear Aspar Mahindra Team | 21 | 18 | 2 | 6 | 1 | 0 | 145 | 4th |
| 2017 | Moto2 | Kalex | SKY Racing Team VR46 | 42 | 18 | 0 | 4 | 0 | 0 | 174 | 5th |
| 2018 | Moto2 | Kalex | SKY Racing Team VR46 | 42 | 18 | 8 | 12 | 6 | 3 | 306 | 1st |
| 2019 | MotoGP | Ducati | Alma Pramac Racing | 63 | 3 | 0 | 0 | 0 | 0 | 9* | 14th* |
| Total | | | | 108 | 10 | 23 | 7 | 5 | 760 | |

**Sentence**

After 4 seasons in the Moto3 category, Francesco Bagnaia moved up to Moto2, racing for SKY Racing Team VR46, where he last rode in 2014.

Figure 7: An example of numerical and temporal reasoning in the above table.

**Trey Johnson**

**Section Title**: College
**Table Section Text**: *None*

| Year | Team | GP | GS | MPG | FG% | 3P% | FT% | RPG | APG | SPG | BPG | PPG |
|------|------|----|----|-----|-----|-----|-----|-----|-----|-----|-----|-----|
| 2003-04 | Alcorn State | 29 | | 28.8 | .414 | .415 | .781 | 2.4 | 2.7 | .5 | .1 | 11.0 |
| 2005-06 | Jackson State | 32 | 29 | 34.8 | .455 | .441 | .750 | 4.8 | 2.3 | 1.4 | .1 | 23.5 |
| 2006-07 | Jackson State | 35 | 35 | 37.8 | .414 | .333 | .743 | 4.5 | 2.6 | .9 | .1 | 27.1 |
| Career | | 96 | | 36.4 | .429 | .382 | .751 | 4.0 | 2.5 | .8 | .1 | 21.0 |

**Sentence**

Trey Johnson averaged 23.5 points, 4.8 rebounds, and 2.3 assists in 32 games.

Figure 8: An example of descriptive sentence in the above table.

3e-6, 5e-6, 1e-5. We used Quadro RTX 8000 GPU for our experiments. We train all our models for 10 epochs, which takes 3 hours for ToTTo and 1.5

| Category | Table | TAPEX NO TAGS | TAPEX RETAG |
|---|---|---|---|
| Numerical, Commonsense, Temporal | Title: Jessica Alba | Spouse(s): Cash Warren ( m. 2008) | Occupation: Actress, businesswoman | Years active: 1994-present . . . | Jessica Alba has been married to Cash Warren since 2008. | Jessica Alba has been active in her profession for over two decades. |
| Commonsense, Temporal | Title: David Foster | Genres: Pop, pop rock, classical, gospel, R&B | Occupation(s): Music executive, record producer, musician, composer, songwriter, arranger | Years active: 1971-present . . . . | David Foster has been performing for over forty years. | David Foster has been performing since the 1970s. |
| Numerical | Title: Dead Poets Society | Directed by: Peter Weir | Running time: 128 minutes | Country: United States | Language: English | Budget: $16.4 million | Box office: $235.9 million . . . . | Dead Poems Society is an American Hollywood film. | Dead Poems Society made over 200 million dollars at the box office. |
| Commmonsense | Title: Hindi Medium | Directed by: Saket Chaudhary | Release date: 19 May 2017 ( 2017-05-19 ) (India), 4 April 2018 ( 2018-04-04 ) (China) . . . . | Hindi Medium was released in the year of its release. | Hindi Medium was released in the summer of 2017. |

Table 8: Generated examples for some samples from the InfoTabs dataset. The predictions generated for TAPEX RETAG were controlled by the corresponding categories listed in the left-most column.

hours for InfoTabs. The TAPEX and T5 family models have 406M and 768M parameters, respectively. The codebooks in RETAG contribute to the additional 3M parameters.

## D  Evaluation Metrics

We used the sacreBLEU implementation (Post, 2018) for computing BLEU scores. For ToTTo, we used the originally released code by authors for computing the PARENT metric. For InfoTabs, we used a lambda weight of 0.1 for computing the PARENT metric.

## E  Example of Generations

We show some instances of generated analytical sentences for the InfoTabs dataset in Table 8. We show generations from the TAPEX without tags model and our proposed TAPEX RETAG model in the table. In the first example, given the numerical, temporal reasoning categories, TAPEX RETAG is able to infer that "*she has been acting for over 20 years*" and "*is still active*" through commonsense reasoning.

## F  Results on Turning Tables

Turning tables (Yoran et al., 2022) is a table based QA dataset with reasoning categories annotated. We observe that our TAPEX RETAG approach outperforms TAPEX for this task as well, when we use the reasoning categories mentioned in their work.

| Dataset | Model | EM | F1 | Overall B1 | B4 | R-L |
|---|---|---|---|---|---|---|
| Turning Tables | TAPEX | 46.49 | 52.61 | 42.06 | 34.00 | 52.62 |
| | + RETAG | **47.44** | **53.52** | **42.97** | **34.74** | **53.12** |

Table 9: Automatic evaluation results for question answering on Turning Tables dataset. Scores are shown in % scale.

## G  Extended Related Work

### G.1  Table-Aware Pretraining

With the success of pre-trained language models on unstructured text, a large number of works have been introduced to incorporate table structure using similar pre-trained strategies. (Herzig et al., 2020; Andrejczuk et al., 2022; Xing and Wan, 2021,?; Yin et al., 2020) introduce a pretraining strategy with specialized objectives for structured data: such as Masked Column Prediction(MCP), Adjacent Cell Prediction (ACP) and so on. Some of the above works, also use special row and column embedding to encode the table structure in the input. (Liu et al., 2021) learns table structure using the task of neural SQL execution over tables in their pretraining strategy. (Dong et al., 2022) presents an elaborate survey for table based pretraining strategies. For any table-specific reasoning task, understanding table structure forms the basis, therefore in this work we use TAPEX (Liu et al., 2021) as our base model. To the best of our knowledge it is the state of the art for ToTTo dataset.

### G.2  Datasets

Newer dataset encompasses (Chen et al., 2020c): consists of diversified sentences across symbolic operations (such as max, min, etc.); (Chen et al., 2020d; Zhu et al., 2021; Chen et al., 2021; Zhao et al., 2022a) introduce table and text based QA tasks that primarily involve numerical and table reasoning along with interactions between table and given text to predict the correct answer. (Nan et al., 2022) is Table-QA dataset that involves complex reasoning. (Suadaa et al., 2021; Chen et al., 2020f, 2019) also consist of references that incorporate numerical and table reasoning.

### G.3 Controllability in Structured Data

There are various works (Parikh et al., 2020; Su et al., 2021b; Wang et al., 2022) that focus on the controlling content while generating from structured data. (Li et al., 2021) uses a prefix set of tokens to better control the topic of the generated text. (Su et al., 2021a) extracts free-form prototypes from a large knowledge base to control the structural formation of the generated text. One of the recent works. To the best of our knowledge, ours is one of the first works to explicitly model for control on various reasoning aspects for structured data to text generation.