# OpenReview forum: "ReTAG: Reasoning Aware Table to Analytic Text Generation"
_EMNLP/2023/Conference — EMNLP 2023 Main_

### Official Review · Reviewer_fnkP · 2023-08-03

**Soundness:** 4

**Excitement:**

3: Ambivalent: It has merits (e.g., it reports state-of-the-art results, the idea is nice), but there are key weaknesses (e.g., it describes incremental work), and it can significantly benefit from another round of revision. However, I won't object to accepting it if my co-reviewers champion it.

**Paper Topic And Main Contributions:**

Most table summarization models use information contained in the table to generate a summary. This paper proposes ReTAG that utilizes multiple categories of reasoning for table summary generation. It shows that with an explicit control of the reasoning categories, the proposed model improves the quality of table summaries based on automatic and human evaluations.

**Questions For The Authors:**

Question A: The claim in the abstract that "While significant progress has been made in table to text generation techniques, models still mostly use information contained within the table to generate the summary." Your problem statement in Section 3.2 is also to generate a summary also using all the information contained in C{ij}, rather than any new information that is not contained within the table. The extra component of your problem statement is that the summary needs to confirm to a subset of R. However, in your introduction section and also Tables 5 & 6, your contributions seem to include an explicit control to pick the reasoning categories. So the selection of the subset R itself is important and should be a task in the problem statement?

Question B: In Abstract: RETAG achieves 1%, 2.7% improvement on the PARENT metric in the relevant slice of ToTTo and InfoTabs for the table to text generation task over state of the art baselines. In Section 1: RETAG outperforms state-of-the-art table to text models on the ToTTo and InfoTabs by 1.8% and 2.2% in BLEU respectively. This information can be found from Table 1 anyway. Is there any reason why different metrics are used to highlight the contributions in different sections?

Question C: Although there are 14 annotators, each instance is labelled by 3 annotators and you only keep those labels voted by at least two raters. Since each label has a score (e.g., partially 0.5), why is it better to keep those labels voted by at least two raters instead of keeping all but averaging their scores? Or do you mean those annotated categories voted by at least two raters?

**Reasons To Accept:**

* Using multiple reasoning categories (with an explicit control of the categories) improves the quality of table summaries based on automatic and human evaluations.

* Approximately 32K instances of reasoning category tagged, table-to-text data (tagged ToTTo & InfoTabs) will be released.

**Reasons To Reject:**

* The problem statement does not appear to be entirely clear with respect to the description in the abstract (See Question A below).

* 1% and 2.7% improvements over the baselines seem insignificant, though 12% improvement is observed based on human evaluation.

* Typos/grammatical issues and inconsistencies make the paper not easy to read.

**Reproducibility:**

3: Could reproduce the results with some difficulty. The settings of parameters are underspecified or subjectively determined; the training/evaluation data are not widely available.

**Reviewer Confidence:**

3: Pretty sure, but there's a chance I missed something. Although I have a good feel for this area in general, I did not carefully check the paper's details, e.g., the math, experimental design, or novelty.

**Typos Grammar Style And Presentation Improvements:**

* There are many typos/grammatical issues throughout the paper  (e.g., "multiple types of reasoning ARE needed", "Refer TO Table 1", "at least", those labels").

* Inconsistent hyphenations (e.g., reasoning-aware vs reasoning aware) and casing for terms (e.g., Table to Text vs table to text).

---

> ### Author Rebuttal · Authors · 2023-08-29
>
> We  thank the reviewer for their appreciation of the strengths of the paper and for the detailed and constructive comments that we address below.
>
> We were also able to increase the annotated ToTTo corpus from 21.7K to 37K sentences. Hence the total dataset size that we will release will be 47K sentences.
>
>
>
> **Reject 1 and Question A about Section 3.2:**
>
>
> We agree with the reviewer that the problem statement can be clarified further in both the abstract and Section 3.2. The clarification we would like to add is that.. “ The task of reasoning-aware table to text is to generate a summary that uses information contained i {Cij}, combine it with world & common sense knowledge and produce a truthful summary S conforming to r”
>
> We also respectfully submit that the task of predicting {R} is while an important problem but is hard to control for in a summarization setting. For instance, in the original ToTTo paper, the references were produced by modifying Wikipedia sentences and hence could have used any reasoning category. In other words, the set of valid reasoning categories that can be used to summarize a table can be very large and hard to determine apriori. To confirm this we also built a classifier to predict the reasoning category for ToTTo and as expected  the task was quite difficult: weighted F1: 53.91%, exact match: 49.14% for multi-label classification in ToTTo.
>
> However, when {R} has to be predicted for QA dataset where R can be predicted from the question, we were able to build a classifier for the Turning Tables QA dataset (Appendix-F in the paper). For Turning Tables the classification performance is much higher at 88% F1.
>
> **Reject 2 and Question B:**
> No, there was not any specific reason to highlight the different metrics in the different sections. The metrics are equally important. This was an oversight and we will highlight them both in the abstract.
>
> We respectfully submit that 1% and 2.7% improvement in PARENT on the well known ToTTo and InfoTabs benchmarks are not an insignificant improvement [1]. This observation also correlates with the significant BLEU (2 points +) improvement and significant improvement on human eval (12%).  We also have added two new baselines (please refer response to Reviewer  CsWy and 5oq8) and the performance improvement of RETAG is consistent.
>
> [1]: https://arxiv.org/abs/2004.14373
>
> **Question C:** The partial score of 0.5 is used only for the human evaluation of model generated outputs. For example, if the gold reasoning categories are {Numerical, Commonsense} and the model generated output has only Numerical reasoning then a partial score of 0.5 is given.
>
> During the main annotation process, we indeed kept the categories voted by at least two raters. Thus, in the above example the gold categories are {Numerical, Commonsense} implies that at least two raters voted Numerical and at least two raters voted Commonsense.

---

### Official Review · Reviewer_5oq8 · 2023-08-05

**Typos Grammar Style And Presentation Improvements:** Please see the reasons to reject sect…
**Soundness:** 4

**Excitement:**

3: Ambivalent: It has merits (e.g., it reports state-of-the-art results, the idea is nice), but there are key weaknesses (e.g., it describes incremental work), and it can significantly benefit from another round of revision. However, I won't object to accepting it if my co-reviewers champion it.

**Paper Topic And Main Contributions:**

This paper proposes methods for improving table-to-text model performance by adding a reasoning-aware module to the baseline model. With this objective, they

(1) introduce a controllable table-to-text dataset by adding an output format requirement (e.g., Temporal Reasoning) as part of the input to the existing dataset;

(2) collect human annotations for categorizing the table summaries into different descriptive and analytical types;

(3) propose the idea of modifying the encoded input latent representation with codebooks for each of the reasoning types defined in this work;

(4) empirically demonstrate that their model can outperform the baseline models on the original table-to-text datasets as well as their modified datasets.

**Questions For The Authors:**

There are a series of T5 and FLAN T5 models with different sizes. Have you compared RETAG with a larger T5 and FLAN T5?

**Reasons To Accept:**

1. This paper studies the controllable data-to-text generation task, which can be a valuable contribution to the standard task setting.
2. They collect a human-annotated dataset for categorizing the table summaries into different descriptive and analytical types and will publically release the dataset.
3. The experimental results suggest the effectiveness of their proposed methods.

**Reasons To Reject:**

1. The experimental setting is not comprehensive enough to support the claim of this paper. Specifically,
-  The baseline models they used for comparison are either smaller or of the same size, which can lead to unfair comparison.
-  A similar concern is that the model proposed in this work (RETAG) is pre-trained on multiple related datasets and fine-tuned on the specific input format (i.e., table summary format requirement + linearized table) proposed in this work. However, while the author did say that the baseline models (not including TAPEX) are fine-tuned on the datasets used in the evaluation, there is still a discrepancy between the training process of RETAG and baseline models because (a) the baseline models are not pre-trained on the related datasets as RETAG is; (b) the authors did not clearly state whether the baseline models are fine-tuned on the same modified input format.
- Following the previous point, the authors did not fine-tune TAPEX using the same process they used for fine-tuning RETAG, while this could have been the most fair comparison.
- The inter-annotator agreement of the human evaluation conducted in this paper is missing.

2. Personly I think the writing of this paper could have been improved. I found it very difficult to follow this paper and there are multiple formatting/writing issues that constantly increase the difficulty of understanding this paper. Just to give a few examples
- In line 110 the authors used the terminology "Tabular Summarization", while in line 052 they used "Tabular Reasoning" (I think they meant the same thing)
- It is probably better to format the content from line 219 to line 225 as a table.
- The notation of Eq.1 and Eq.2 is confusing. The authors seem to be overloading the term Q^r().

**Reproducibility:**

3: Could reproduce the results with some difficulty. The settings of parameters are underspecified or subjectively determined; the training/evaluation data are not widely available.

**Reviewer Confidence:**

3: Pretty sure, but there's a chance I missed something. Although I have a good feel for this area in general, I did not carefully check the paper's details, e.g., the math, experimental design, or novelty.

---

> ### Author Rebuttal · Authors · 2023-08-29
>
> We  thank the reviewer for their appreciation of the strengths of the paper and for the detailed and constructive comments that we address below.
>
> We were also able to increase the annotated ToTTo corpus from 21.7K to 37K sentences. Hence the total dataset size that we will release will be 47K sentences.
>
> **Reject-1a:**
> * We first clarify that all baselines (including TAPEX) were fine tuned exactly like how ReTAG was fine tuned. Specifically the datasets used to fine tune (ToTTo, InfoTabs) and the data format (linearized table and reasoning  tags) was identical as ReTAG.
> * However we acknowledge that the suggested ablation experiment could be very helpful. In this experiment, we retrain TAPEX with exactly the same pre-training datasets, fine tuning datasets and format as used for ReTAG. Therefore the only difference here would be the vector quantized reasoning codebooks and the accompanying loss function.  Specifically,  we took the TAPEX checkpoint and pre-trained it on the same reasoning corpora with reasoning tags using an identical training process that we used in ReTAG. We then fine-tuned the reasoning enhanced TAPEX model on the ToTTo and InfoTabs dataset with reasoning tags in the inputs. We found that ReTAG outperforms this model on both the datasets. The full results are as follows:
>
> | Dataset  | Model                | Overall | Overall | Analytical | Analytical | Descriptive | Descriptive |
> |----------|----------------------|:-------:|:-------:|:----------:|:----------:|:-----------:|:-----------:|
> |          |                      |   BLEU  |  PARENT |    BLEU    |   PARENT   |     BLEU    |    PARENT   |
> | ToTTo    | TAPEX-Tagged with Pre Training |  53.83  |  53.15  |    46.99   |    44.42   |    55.11    |    53.79    |
> | ToTTo    | ReTAG with Pre Training        |  55.18  |  54.37  |    48.32   |    45.84   |    56.45    |    55.72    |
> | InfoTabs | TAPEX-Tagged with Pre Training |  37.98  |  34.95  |    34.78   |    23.41   |    38.09    |    43.20     |
> | InfoTabs | ReTAG with Pre Training        |  38.82  |  36.58  |    35.19   |    24.01   |    40.90    |    45.36    |
>
>
> Note that the ReTAG model without pre-training also outperforms the TAPEX model without pre-training. The results for ReTAG without pre-training are already provided in the paper in Table 3. We will compile the results in a single table for easier comparison. We chose TAPEX in the rebuttal since it has the best performance among the chosen baselines. We are happy to report numbers of this ablation experiment for the other 3 baselines as well.
>
> **Reject-1b**
>
> The Fleiss' kappa score between the three annotators for human evaluation are as follows:
> Reasoning: 0.5011
> Faithfulness: 0.5583
> Coverage of Highlighted Cells: 0.6722
> These show very good inter rater agreement for the human evaluation task.
>
> **Reject 2:**
> Thanks for pointing them out. We will improve the writing and the presentations of the paper to make the draft more clear.
>
> **Question 1:**
> We found that the performance of large versions of FLAN-T5 are as follows:
>
> | Dataset  | Model                | Overall | Overall | Analytical | Analytical | Descriptive | Descriptive |
> |----------|----------------------|:-------:|:-------:|:----------:|:----------:|:-----------:|:-----------:|
> |          |                      |   BLEU  |  PARENT |    BLEU    |   PARENT   |     BLEU    |    PARENT   |
> | ToTTo    | FLAN-T5 Large UnTagged  |  54.67  |  53.75  |    47.74   |    45.13   |    56.38    |    55.81    |
> | ToTTo    | FLAN-T5 Large Tagged |  55.31  |  54.11  |    47.86   |    45.41   |    56.97    |    56.12    |
> | ToTTo    | ReTAG        |  55.18  |  54.37  |    48.32   |    45.84   |    56.45    |    55.72    |
> | InfoTabs | FLAN-T5 Large UnTagged  |  36.47  |  34.92  |    32.61   |    23.79   |    38.27    |    43.98    |
> | InfoTabs | FLAN-T5 Large Tagged |  37.72  |  35.43  |    34.65   |    24.38   |    39.14    |    44.77    |
> | InfoTabs | ReTAG        |  38.82  |  36.58  |    35.19   |    24.01   |    40.90    |    45.36    |
>
> The FLAN T5-Large model has 783M parameters which is nearly two times of ReTAG, which has 407M parameters. Even with half the number of parameters, ReTAG is able to almost reach the same performance as FLAN T5 for ToTTo and for InfoTabs  even surpass it. For ToTTo, RETAG is able to outperform FLAN-T5 Large on the analytical slice. Combining this observation with the previous ablation on effectiveness of vector quantization, we expect the result to improve with larger models. We are happy to train larger FLAN-T5 checkpoints with our codebook architecture in the camera ready version, if accepted.
>
> Also as part of response to reviewer CsWy, we have reported performance with a 24 Billion instruction-tuned language model. We show that RETAG is able to outperform the LLM 24B in a few shot setting.
>
> Also, we would like to point out that the usage of tags (reasoning awareness in Contribution-1) helps improve the performance of FLAN T5 tagged over FLAN-T5 untagged.  Hence this experiment also shows that reasoning awareness can improve the performance of table summarization models as noted by the reviewers in reasons to accept.

---

### Official Review · Reviewer_CsWy · 2023-08-05

**Soundness:** 4

**Excitement:**

4: Strong: This paper deepens the understanding of some phenomenon or lowers the barriers to an existing research direction.

**Missing References:**

Feng Nie, Jinpeng Wang, Jin-Ge Yao, Rong Pan, and Chin-Yew Lin. 2018. Operation-guided Neural Networks for High Fidelity Data-To-Text Generation. In Proceedings of the 2018 Conference on Empirical Methods in Natural Language Processing, pages 3879–3889, Brussels, Belgium. Association for Computational Linguistics.

This paper also discuss reasoning over data-to-text task, which should be discussed in the related work.

**Paper Topic And Main Contributions:**

This paper addresses the problem of table summarization with multiple types of reasoning and access to knowledge beyond the table's scope.

The main contributions of this paper are:

1. The introduction of RETAG, a vector-quantized encoder-decoder table-to-text model with explicit control across reasoning categories.

2. Through human evaluation, the authors show that RETAG's output is up to 12% more faithful and analytical compared to a strong table-aware model.

3. The extension and open-sourcing of 32K tagged instances in the ToTTo and InfoTabs datasets with the reasoning categories used in each reference.

**Questions For The Authors:**

- Did you consider use ChatGPT to aid the data construction (e.g., to annotate the reasoning category)?

**Reasons To Accept:**

1. The paper addresses an important gap in table summarization by incorporating multiple types of reasoning and access to knowledge beyond the table, which is a novel approach.
2. The human evaluation results show that RETAG's output is up to 12% more faithful and analytical compared to a strong table-aware model, indicating the practical usefulness of the approach.
3. The authors extend and open-source 32K tagged instances of the ToTTo and InfoTabs datasets with reasoning categories used in each reference, which is a valuable contribution to the research community.

**Reasons To Reject:**

- This paper does not compare with current ChatGPT models with appropriate reasoning prompting commands, which may be a more simple method for current data-to-text generation task.

**Reproducibility:**

4: Could mostly reproduce the results, but there may be some variation because of sample variance or minor variations in their interpretation of the protocol or method.

**Reviewer Confidence:**

4: Quite sure. I tried to check the important points carefully. It's unlikely, though conceivable, that I missed something that should affect my ratings.

---

> ### Author Rebuttal · Authors · 2023-08-29
>
> Thank you for your positive review of our work. Please find the responses below:
>
> **Reject 1 and Question 1:**
>
> We tried experimented with a large 24B  instruction-tuned large language model (similar to chatGPT )  in a few-shot setting and found that they have difficulties understanding the table structure and processing long tables (commonly found in ToTTo). The results are as follows:
>
> | Dataset  | Model                | Overall | Overall | Analytical | Analytical | Descriptive | Descriptive |
> |----------|----------------------|:-------:|:-------:|:----------:|:----------:|:-----------:|:-----------:|
> |          |                      |   BLEU  |  PARENT |    BLEU    |   PARENT   |     BLEU    |    PARENT   |
> | ToTTo    | Instruction LLM (24B)|  38.73  |  35.17  |    34.15   |   30.96    |    39.76    |    37.29    |
> | ToTTo    | ReTAG with PreTraining        |  55.18  |  54.37  |    48.32   |    45.84   |    56.45    |    55.72    |
>
> We use the following few shot prompt for the LLM:
>
> ```bash
> You are proficient in understanding tables.Given the highlighted cells in the table and reasoning category, generate a summary focusing on the highlighted cells using the given reasoning category.
>
> If no reasoning category is given, generate a descriptive sentence. The table is given as a sequence of tokens arranged row wise. The highlighted cells are marked with [hl].
>
> Input: title: julius erving; section: regular season; col : year | team | gp | gs | mpg | fg% | 3p% | ft% | rpg | apg | spg | bpg | ppg row 1 : [hl] 1972–73 [hl] | [hl] virginia (aba) [hl] | 71 | nan | 42.2 | 0.496 | 0.208 | 0.776 | 12.2 | 4.2 | 2.5 | 1.8 | [hl] 31.9 [hl]
> Reasoning Category: No Reasoning Category
> Summary: in the aba, julius erving averaged 31.9 points per game in the 1972–1973 season.
>
> Input: title: simon williams (athlete); section: international competitions; col : year representing the great britain and england | competition representing the great britain and england | venue representing the great britain and england | position representing the great britain and england | event representing the great britain and england | notes representing the great britain and england row 1 : 1992 | olympic games | barcelona, spain | 28th (q) | discus throw | [hl] 53.12 m [hl]
> Reasoning Category: numerical, table, external knowledge
> Summary: simon williams was well down on his best there, however, managing only 53.12 m in the qualifying round.
> ```

---

### Meta-Review · Area_Chair_f7at · 2023-09-23

**Recommendation:** 3

**Metareview:**

The authors present a new table-to-text generation dataset along with a model and a credible evaluations. Their human annotator results are well constructed with reasonable reliability. The authors also seem to have allayed some of the concerns from reviewers in their rebuttal.

I encourage the authors to follow reviewer  5oq8's advice and suggestions to increase the clarity of writing in the paper. There's

---

### Decision · Program_Chairs · 2023-10-07

**Decision:**

Accept-Main

**Comment:**

The authors present a new table-to-text generation dataset along with a model and a credible evaluations. Their human annotator results are well constructed with reasonable reliability. The authors also seem to have allayed some of the concerns from reviewers in their rebuttal.

I encourage the authors to follow reviewer  5oq8's advice and suggestions to increase the clarity of writing in the paper. There's